# Thermoregulatory and Feeding Behavior under Different Management and Heat Stress Conditions in Heifer Water Buffalo (*Bubalus bubalis*) in the Tropics

**DOI:** 10.3390/ani11041162

**Published:** 2021-04-18

**Authors:** Maykel Andrés Galloso-Hernández, Mildrey Soca-Pérez, Devon Dublin, Carlos Armando Alvarez-Díaz, Jesús Iglesias-Gómez, Cipriano Díaz-Gaona, Vicente Rodríguez-Estévez

**Affiliations:** 1Department of Animal Production, Veterinary of Faculty, Córdoba University, 14071 Córdoba, Spain; pa2digac@uco.es (C.D.-G.); pa2roesv@uco.es (V.R.-E.); 2Experimental Station: “Indio Hatuey”, Central España Republicana, Matanzas 44280, Cuba; mildrey@ihatuey.cu (M.S.-P.); iglesias@ihatuey.cu (J.I.-G.); 3Language Department, Hokkaido University of Education, Program Advisor, Sapporo 060-8611, Japan; devdub@yahoo.com; 4Academic Unit of Agrarian Science (UACA), University Technician of Machala, El Oro 070211, Ecuador; arma25cu@yahoo.com

**Keywords:** feeding behavior, grazing, browsing, wallowing, animal welfare, silvopastoral system, heat stress

## Abstract

**Simple Summary:**

Silvopastoral systems can modulate the thermoregulatory behavior of buffaloes decreasing the heat stress and improving the animal welfare in the tropics. The objective of this study was to compare the behavior of heifer buffaloes in a silvopastoral systems with *Leucahena leucocephala* trees and a conventional system without trees under two heat stress condition (intense heat stress and moderate heat stress) in Cuba. The results show that despite intense heat stress conditions, the animals spent more time feeding in the silvopastoral system than in the conventional system. Besides, the silvopastoral system reduced the use of water in the wallowing areas. We conclude that pastures with trees increase fodder offer while improve grazing behavior and animal welfare for buffalo farming in tropical conditions.

**Abstract:**

In the wake of climate change and global warming, the production systems of water buffaloes (*Bubalus bubalis*) are receiving increasing attention in the tropics, where the silvopastoral systems can improve animal welfare and production conditions. The objective of this study was to characterize the behavior of heifer buffaloes in a silvopastoral system (SPS) with *Leucaena leucocephala* (600 trees/ha) and in a conventional system (CVS), under intense heat stress and moderate heat stress in Cuba. We observed nine animals, with an average weight of 167.9 kg at the beginning of the study, during the daylight period, from 6:00 to 18:00 h, at 10 min intervals, for 12 days. Activities recorded were grazing, ingestion of tree leaves, rumination, water intake, walking, lying, standing, sheltering in the shade of trees, and wallowing. Sheltering in the shade of trees and wallowing were collectively considered as thermoregulatory behavior (TB). TB was different in both systems and conditions of heat stress (*p* < 0.05), with 4.06 in CVS and 3.81 h in SPS in the intense heat stress period, while it was 2.91 and 1.08 h for SPS and CVS, respectively, during the moderate heat stress period. The wallowing activity showed statistically significant differences (*p* < 0.05) in the intense heat stress season with 1.18 and 2.35 h for SPS and CVS, respectively. Time spent on feeding behavior was highest in the SPS system (*p* < 0.05). Longer times of thermoregulatory and feeding behavior indicate the importance of trees in animal welfare for this species in tropical conditions, thus supporting avoided deforestation and the replanting of trees in existing production systems and landscapes.

## 1. Introduction

The water buffalo (*Bubalus bubalis*) is a species that has been recently introduced in Cuba [1,2,3] and other tropical countries in the Americas such as Brazil and Colombia [4], where they are managed in grazing systems with the provision of shade [4,5,6].

With the increase in solar radiation, and the increase in temperature because of climate change [7], animals are forced to change their habits during the daylight period, when the stressors of climate have a higher influence on their behavior [8]. Heat stress affects animal behavior, causes production losses, and can impair animal welfare.

Alternatives such as the use of trees, changes in management and grazing schedules, and the genetic improvement of species more in line with the new climatic conditions are some of the strategies considered to address this issue [9].

Housing systems and pastoral systems with different forest arrangements for buffaloes are known in Latin American countries [10,11,12]. For example, buffaloes housed with artificial shade and water sprinklers for cooling in a monocropping natural grass system are observed in the wallowing area to release excess heat [5,6]. In this regard, Simon and Galloso [6] found a better milk production per hectare and a better daily weight gain of buffalos reared in silvopastoral systems (SPSs) in comparison with conventional systems (CVSs). Recently, a higher daily weight gain was reported for buffalos compared to bovine males during the growing phase in a silvopastoral system [2].

However, the effect of trees on the grazing behavior of buffaloes in hot-humid conditions has not been studied in depth, and it is not known whether the presence of trees can help their thermoregulatory behavior and increase grazing activity while simultaneously reducing wallowing time.

Hence, the objective of this research was to characterize foraging activity and thermoregulatory behavior of heifer buffaloes in the tropics in SPSs and CVSs, comparing both in the intense heat stress and moderate heat stress conditions in Cuba.

## 2. Materials and Methods

### 2.1. Study Site and Animals

The study was conducted in Périco, Matanzas, Cuba, located at 22°48′7″ latitude north and 81°1′ longitude west (19.01 msm), in hydrated red ferralitic soil under climatic conditions [2] that were previously described by Galloso-Hernández et al. [11]. Buffaloes used were heifers of 12–18 months of age with an average weight of 167.9 kg (at the beginning of the experimental period) and subjected to the management regime outlined by Galloso-Hernández et al. [11]. The animals engaged in grazing pasture during the day and were taken to a paddock at night. In the grazing area, they had access to a wallowing area, natural shade under trees (*Dichrostachys cinerea*), while drinking water and mineral salts were provided ad libitum. 

### 2.2. Experimental Design and Description of Production Systems

We studied the influence of different heat stress conditions on the thermoregulatory and feeding behavior of heifer buffaloes in two production systems. The experimental design followed a longitudinal analysis method in the same group of animals under four different experimental conditions previously described [11] and shown in Table 1. This method was used because it sought to minimize individual variability, reduce agonistic behaviors, and monitor the influence of experimental conditions over time.

Measurements were made in 12 h day cycles for three consecutive days in each experimental condition (T1, T2, T3, and T4), totaling 12 days. Animals were first placed in Experimental Condition 1 followed by Experimental Conditions 4, 2, and 3, respectively. Between Treatments 1 and 4 and between Treatments 2 and 3, there was an adaptation time of 15 days where they were subjected to the same management and adaptation recommended by Martyn and Bateson [12]. This adaptation period ensured that the previous experimental condition did not influence the next experimental condition to which they were subjected. The experimental area of study (i.e., conventional system (CVS) only pastures, Figure 1A) and the silvopastoral system (SPS)(i.e., pastures combined with trees, Figure 1B); under two conditions of heat stress, moderate (THI < 75) and intense (THI> 75 is shown in Figure 1.

### 2.3. Temperature and Humidity Index

We measured the temperature-humidity index (THI) [13], which was calculated using the formula THI = (1.8 × T + 32) − [(0.55 − 0.0055 × RH) × (1.8 × T − 26)], where T is the air temperature (°C) and RH is the relative humidity (%). 

We determined heat stress to be intense (THI > 75) and moderate (THI < 75), as was reported by Pérez et al. [8] when describing the conditions of intense heat stress in Cuba (from May to October) and the conditions of moderate heat stress (from November to April). The experimental measurements were made at the beginning of each season, in the months of May and November, respectively (Table 1). The environmental temperature and relative humidity of the site was measured for each system under moderate heat stress and intense heat stress at the height of the withers in animals, as was reported previously [11].

### 2.4. Behavioral Observations

Feeding and thermoregulatory behaviors of buffaloes were measured using the direct observation method [11,14]. The number of animals for each activity between each measurement interval was recorded. The time spent was calculated based on the application of the equation as proposed by [14,15]. Each observation cycle consisted of 72 observations made over three days with a 10 min interval between observations from 6:00 to 18:00 h (Table 1). We then grouped the related variables. Active feeding behavior was calculated as the sum of grazing and browsing. Feeding behavior was determined as the sum of active grazing behavior, rumination, and water consumption. Thermoregulatory behavior was considered as the sum of wallowing and shading behavior (Table 2).

### 2.5. Statistical Analysis

The SPSS^®^ software version 25 was used for statistical analysis (IBM Corp^®,^ accessed on 15 April 2021). An analysis of variance (ANOVA) was applied to find the differences between the behaviors, considering the levels of intense and moderate heat stress and the type of system (silvopastoral or conventional). We determined the analysis of variance after checking the distribution of normality of the times dedicated to each activity by utilizing the Kolmogorov–Smirnov test. The average time dedicated to each activity was compared by stress levels and systems with Duncan’s multiple range comparison test [16] in order to detect the inequalities between the means.

## 3. Results

### 3.1. Environmental Conditions

During the moderate heat stress period, temperature (T) and relative humidity (RH) did not show significant differences between the CVS and the SPS. However, in the intense heat stress period, when the values of temperature increased exponentially as daylight hours advanced, temperature was always 2 °C lower in the SPS around midday (Table 3).

The RH was higher in the SPS in the intense heat season. According to Penton and Blanco [17], this is explained because this humidity is introduced into the system by the tree cover and the effect of accumulative humidity in the grass.

### 3.2. Grazing and Feeding Behaviors

The highest and the lowest fodder availability for the SPS during the intense heat stress season and the CVS during the moderate heat stress season were 6.68 and 2.19 Ton (DM) Dry Matter/(ha) hectare and per rotation, respectively (Table 4). Moreover, the highest and the lowest grazing pressures during the intense heat stress period for the SPS and CVS were 23.07 and 8.52 Kg DM/100 kg BW, respectively.

The time spent on each activity is shown in Table 5 and Table 6. During the intense heat stress season, the water intake was higher in the SPS than in the CVS with 0.35 vs. 0.07 h, respectively (Table 5).

During the intense heat stress period, there were significant statistical differences for the ingestion of leaves from trees; this was higher in the SPS than in the CVS (*p* < 0.05), with 0.31 vs. 0.21 h dedicated to this activity, respectively. During the moderate heat stress period, the animals dedicated 0.71 vs. 0.10 h for the ingestion of leaves from trees in the SPS and the CVS, respectively (*p* < 0.05) (Table 5). Ingestion of leaves in the CVS basically was executed in the wallowing plot browsing *D. cinerea*.

The feeding behavior showed significant differences between moderate and intense heat stress condition and systems (*p* < 0.05), with the most time spent on these activities in the SPS during the intense heat stress (10.47 h) and the least time in the SPS during the moderate heat stress period (6.84 h) (Table 5).

During the moderate heat stress period, the grazing behavior did not show statistically significant differences between both systems, with an average of 4.82 h. However, this behavior was significantly different during the intense heat stress period, with 7.49 vs. 5.96 h, respectively, in the SPS and the CVS (*p* < 0.05) (Table 5).

### 3.3. Thermoregulatory Behavior and Other Activities

During the intense heat stress period, the animals spent 1.18 vs. 2.35 h on wallowing for the SPS and the CVS systems, respectively (*p* < 0.05). In addition, during this season, the animals were sheltering in the shade of trees during 2.62 vs. 1.71 h in the SPS and in the CVS, respectively (*p* < 0.05) (Table 6).

Sheltering in the shade of trees had remarkable differences in the moderate heat stress condition with 1.99 vs. 0.61 h for the SPS and the CVS, respectively (*p* < 0.05) (Table 6).

The sum of wallowing and sheltering, considered as thermoregulatory behavior, showed significant differences between the systems depending on the heat stress level. Under moderate heat stress, it was 2.91 h vs. 1.08 h in the SPS and the CVS, respectively (*p* < 0.05); in the intense heat stress period, this thermoregulatory was 4.06 vs. 3.81 h for the SPS and the CVS, respectively (*p* < 0.05) (Table 6).

It was found that there were no statistically significant differences between lying down and walking under both heat stress conditions, but figures were higher in the CVS system (Table 6).

During the moderate heat stress period, the animals spent 3.66 vs. 3.06 h standing for the SPS and the CVS, respectively (*p* < 0.05); under the intense heat stress, they spent 2.77 vs. 3.26 h standing for the SPS and the CVS, respectively (*p* < 0.05) (Table 5).

### 3.4. Correlations between Grouping Variables of Behaviour

The analysis of correlations of Pearson between grouping variables showed that the grazing and feeding behaviors have significant correlation (r = 0.608; *p* < 0.01), while the feeding and thermoregulatory behaviors showed a high correlation (r = 0.83; *p* < 0.01).

## 4. Discussion

The dry season in Cuba coincides with moderate temperatures, which is favorable for animal welfare; however, a harsher season, with higher temperatures, would have a negative effect on animal welfare [9,11,18,19]. The HR season is the period of the year when animals developed intense heat stress and spent more time in the wallowing area, which is attributed to the higher temperature regime in that period, above 35 °C. Regardless of the expected comfort in the SPS, the animals were motivated to enter the water for immersion, although it is important to consider that the animals also had shade available in the wallowing area, where there were trees around the pond [20]. Frisch and Vercoe [21] indicated that this species has only one-sixth of the number of sweat glands of cattle (*Bos indicus*) and is sparsely covered with hairs. In addition, its dark skin, thick epidermis, and less dense sweat glands make it difficult for these animals to resist high temperatures and a dry environment [22]; hence, buffaloes tend to suffer heat stress when they are exposed to solar radiation [21] and seek water for immersion to avoid this stressful environmental condition [23]. It is important to consider that this species is well adapted to swamps and areas subject to flooding [24].

One of the limitations of this study was the reduced number of animals used, which in turn helped improve the effectiveness of behavioral records by reducing agonistic behaviors [12]. The longitudinal design additionally contributed to the robustness of the subsequent analyses in this study. This contributed to the understanding of the evolution of the behavior of these animals subjected to different conditions of thermal stress under shade and in the CVS, in accordance with what was previously described by Galloso [11].

The differences in time spent in the shade of trees in the moderate heat stress season between the SPS and the CVS are because, during this season, the paddocks had lower grass availability and were dryer, especially in the CVS system; hence, the animals of the SPS tried to satisfy their need for fodder grazing under the trees.

The relation established (animals, wallowing, and tree shade) is more complex than is simply observed when only wallowing is considered. While wallowing allows for cooling with water or mud, this cannot be considered as enough for animal welfare, because radiation in the head and spine is not avoided [11,20]. It would be interesting to have data on how radiation influences the temperature in different areas of the body, as was done under infrared thermography, through which microvascular changes in the head were previously studied, particularly in the area of the eye orbits and in the muscles of the spine, scrotum, or mammary gland [24,25], both while in and out the water, and with or without tree shade [26].

When buffaloes are more affected by stress factors (solar radiation, heat stress, and excess temperatures) [27,28], they prefer to eat in the shade [11,20,27]. This has been described in other species; i.e., in cattle breeds of the tropics where time spent in the shade has been positively correlated with mean radiant temperature and solar radiation [29].

The statistical differences found (thermoregulatory behavior, wallowing, and browsing), particularly in the SPS during the intense heat stress season, have also been reported for grazing cattle in SPSs [2,10].

The differences in sheltering behavior during the intense heat stress season are probably influenced by the low nutritional values of the pastures in the CVS [26,30], which forces the animals to return earlier to the grazing paddock in order to satisfy their nutritional requirements.

Similar results for the thermoregulatory behavior in favor of silvopastoralism were reported by Yadav et al. [31] when comparing the influence of different methods of cooling down on productivity, and metabolic and blood profiles; they found that nebulization or wallowing favors blood indicators. In periods of greater heat stress, the silvopastoral environment reduces the temperature by 1.5 °C in comparison to direct sunlight [28]. Other cooling systems (showers and artificial shade) have been reported by Barros et al. [26] and Sevegnani et al. [29], and, with the use of infrared thermography, it was proven that the buffalo body temperature is reduced by around 2 °C.

An element to be considered on buffalo farms is that trees provide a better environment and assure more welfare in the tropics. Trees have advantages for hygrothermal stability in the farm [28] and food stability [1] and must be considered as important handling alternatives for the welfare of the species [6].

The results found of longer feeding behavior in the SPS could be attributed to the filter shade values, as was found by Penton and Blanco [17], who observed that trees such as *L. leucocephala* and other legume species offer filterable radiation in SPSs in the tropics and reported between 10% and 30% more production of dry matter of grass in SPSs compared with monoculture pastures [30,32].

It is known that SPSs (pastures, trees, and animals) contribute to improving the condition of the grass (more nutritional values) and animal welfare [18,30,33]; this is also the case in other latitudes, species, and agroforestry systems [34].

The time spent ingesting tree leaves was always higher in the SPS, which is quite obvious, although the consumption of tree branches was also possible when the animals were in the wallowing area in both systems. This means an additional effort is required to browse this resource in the CVS, because this fodder was only available there while they wallowed under the *D. cinerea* canopy.

These observations confirm the importance of feeding ruminants with branches or tree fodders as a nutritional complement of the ration. It is known that the ingestion of leaves from trees favors the uptake of nitrogen, and this could be the cause of the increase in its consumption, helping the ruminal microflora, as reported by Wanapat and Phesatcha [35]. Hence, it is important to consider the role of trees in increasing the supply of nitrogen in a diet at times when pastures are not meeting the requirements of animals [36]. Different authors [18,32,33] reported that, when SPSs have high edible biomass availability (higher than 30 Ton DM/ha and year), of which pasture represents 75–90% of the fresh diet intake and tree foliage represents 10–25% of the fresh fodder, animals improve their performance.

*Leucaena* produces between 14.2 and 18.0 Ton DM/ha a year with irrigation (whole plant) and between 7 and 14 Ton DM/ha a year under rain-fed conditions [32,33]. Its contents of DM (24–27%), protein (20–24.26%), DAF (30%), Ca (0.83–2.0%), and P (0.29–0.38%) fluctuate.

*D. cinerea* is an invasive species in Cuba and its nutritional composition is 44.2% DM, 14.7% protein, and 30.3% DNF [37].

The sum of activities included in grazing behavior (grazing and ingestion of tree leaves) showed a significant correlation with the sum of activities included in feeding behavior (grazing, rumination, the ingestion of tree leaves, and water intake). Furthermore, the thermoregulation behavior (wallowing and sheltering in the tree shade) showed an interesting correlation and corroborates how important the thermoregulation activities are in these conditions of intense heat stress.

Hence, tree shade is necessary because wallowing areas alone are not enough. Therefore, these animals need tree shade to complement their cooling needs.

In the CVS, the longer time spent wallowing under intense heat stress conditions, with 2.35 h, and the time positioned under the shade of the *D. cinerea* trees in the wallowing area, with 1.71 h, reflects the greater need for cooling in this season.

In the SPS during the intense heat stress season, the elevated combination of wallowing and shading, 1.18 h and 2.62 h, equivalent at 3.81 h in thermoregulatory behavior, suggests that the animals spend more time in the shade of trees to forage comfortably. However, wallowing does not seem to be as important under moderate heat stress conditions under SPS.

Obviously, there was sheltering under the shade of trees for a longer time in the SPS. Part of this time was spent positioned under the trees in order to browse tree leaves.

The less time spent in the shade of *D. cinerea* in the CVS and the wallowing behavior suggest that the animals replaced the activities of cooling for collecting food in the CVS paddock.

During the diurnal period, grazing is reduced by the influence of stressors, such as high temperature, and modifies some metabolic indicators related to animal welfare, such as thyroid hormones, respiratory rate, and rectal temperature [8,26]. In this regard, the highest index of consumption in the SPS in both stress conditions can be considered proof that grazing time increases when animals are less stressed [8].

The rumination reduction in intense heat stress conditions can be attributed to the better quality in the diet [30], despite a greater availability of fodder (Table 4). Furthermore, in the intense heat stress period, the rumination time increased in the SPS, in contrast to the CVS.

In the SPS, it is possible that an increase in bathing time during the intense heat stress season contributes to a reduction of foraging activities in the afternoon.

In short, these results (regarding time spent wallowing, grazing, and engaging in thermoregulatory behavior in the SPS) can explain the role played by trees distributed in the paddocks and can explain the most productive results found by some researchers in bovine [18,19,32] and in male buffaloes [2] in the tropics, with 0.775 kg/d of daily weight gain and 4.32 Ton DM/ha per rotation during 180 days of growth period for this species in the last studies to which we referred.

These elements can be particularly important to understanding why heifer buffaloes maintain their wallowing habits independently of the provision of natural shade, particularly because, as some authors suggest [6,38,39,40], more attention should be paid to shade to improve animal welfare and performance.

## 5. Conclusions

The feeding behavior of buffaloes is favored in silvopastoral systems, with a special influence on the ingestion of tree leaves. Furthermore, silvopastoralism is an alternative that can improve the management of different heat stress conditions and welfare of buffaloes in the tropics and could reduce the use or need for wallowing areas.

Further studies should determine the influence of shade on the temperature of different parts of the body in silvopastoral systems, and these systems with wallowing areas should be studied to determine whether this species can adapt to dry agroforestry systems. It would be convenient to introduce the use of the black globe temperature–humidity index (BGTHI) and employ infrared thermography cameras to increase the sensitivity in future research measurements.

In any case, everything shows that more attention should be paid to shade for animal welfare and to improve buffalo production in the tropics as a means of counteracting the effects of increased temperature due to climate change.

## Figures and Tables

**Figure 1 animals-11-01162-f001:**
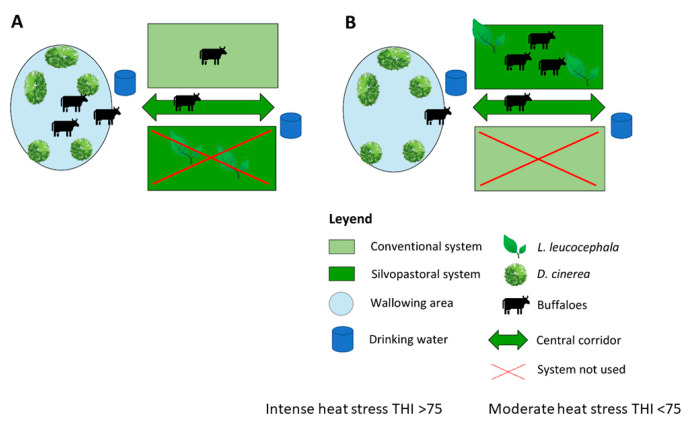
Description and classification of site of study. conventional system (CVS) only pastures, Figure 1**A**, pastures combined with trees, Figure 1**B**.

**Table 1 animals-11-01162-t001:** Outline of experimental conditions.

Treatments	Description	Month of Measurement	Total Number of Observations	Frequency (minutes)	Days of Evaluation	Number of Observation Cycles	Hours of Observation
T1	Conventional system in intense thermal stress	May	875	10	3	216	36
T2	Conventional system in moderate heat stress	November	875	10	3	216	36
T3	Silvopastoral system in moderate thermal stress	May	875	10	3	216	36
T4	Silvopastoral system in intense thermal stress	November	875	10	3	216	36

**Table 2 animals-11-01162-t002:** Description of recorded activities.

Behavioral Activity	Description
**Active feeding behavior**	
Grazing	Time spent eating grass in the paddocks
Browsing	Time spent browsing, i.e., the process of consuming the tips of branches and tree leaves.
**Passive feeding behavior**	
Rumination	Time spent on rumination, i.e., the process of regurgitating previously ingested food and masticating it a second time.
Water intake	Time spent on water intake, i.e., consuming water in the central corridor.
**Thermoregulatory behavior**	
Shading	Performing any activity under the trees (in the systems without trees, it was possible in the wallowing area).
Wallowing	Time spent wallowing, i.e., having a bath in the pond of water to cool.
**Others**	
Walking	Moving from one place to the next
Lying down	Lying, resting, or ruminating in a decubitus position *
Standing	Standing without any other activity or ruminating *

* This activity can be simultaneous with other activities.

**Table 3 animals-11-01162-t003:** Temperature and relative humidity values registered for each system (with or without trees) and season of rain (light or heavy) in the research area.

	Definition	N	Temperature (°C)	Relative Humidity	Temperature–Humidity Index
T1	Conventional system under intense heat stress	875	33.01 (±7.63) ^b^	53.41 (±20.98) ^a^	81.75 (±18.53) ^b^
T2	Conventional system under moderate heat stress	875	29.31 (±12.80) ^a^	51.07 (±25.44) ^a^	76.12 (±16.79) ^a^
T3	Silvopastoral system under moderate heat stress	875	28.97 (±7.78) ^a^	51.63 (±21.95) ^a^	76.40 (±8.61) ^a^
T4	Silvopastoral system under intense heat stress	875	31.00 (±5.86) ^a,b^	59.31 (±18.60) ^b^	79.98 (±6.34) ^b^

^a,b^ Means in the same column with different superscripts differ significantly (*p* < 0.05). N: number of observations.

**Table 4 animals-11-01162-t004:** Fodder offer and grazing pressure.

Treatment	System and Season	Availability of Fodder in Dry Matter/ha and Per Rotation	Grazing Pressure
T1	Conventional system; intense heat stress	4.27 Ton/ha	8.52 Kg DM/100 kg body weight
T2	Conventional system; moderate heat stress	2.19 Ton/ha	14.8 Kg DM/100 kg body weight
T3	Silvopastoral system; moderate heat stress	3.92 Ton/ha	17.16 Kg DM/100 body weight
T4	Silvopastoral system; intense heat stress	6.68 Ton/ha	23.07 Kg DM/100 Kg body weight

DM (Dry matter).

**Table 5 animals-11-01162-t005:** Comparison of the grazing behavior of the water buffaloes depending on the pastoral system and the season: mean number of animals doing the activity (standard error) and time in hours.

Treatment	Definition	Grazing (G)		Rumination (R)		Water Intake (D)		Ingestion of Leaves from Trees (B)		G + B		Feeding behavior (FB) (G + R+D + B)	
		Number of Animals	Time (h)	Number of Animals	Time (h)	Number of Animals	Time (h)	Number of Animals	Time (h)	Number of Animals	Time (h)	Number of Animals	Time (h)
T1	Conventional system under intense heat stress	5.22 ^a^ ± 1.07	4.31 ^a^	2.62 ^b^ ± 0.99	4.40 ^b^	0.33 ^a^ ± 0.24	0.07 ^a^	0.04 ^a^ ± 0.37	0.21 ^a^	5.36 ^b^ ± 0.74	5.96 ^b^	8.17 ^c^ ± 0.44	9.08 ^c^
T2	Conventional system under moderate heat stress	4.97 ^a^ ± 1.18	5.52 ^a^	2.72 ^b^ ± 1.14	3.02 ^b^	0.23 ^a^ ± 0.22	0.25 ^a^	0.10 ^a,b^ ± 0.31	0.1 ^a b^	4.47 ^a^ ± 0.83	4.95 ^a^	6.83 ^b^ ± 0.44	7.61 ^b^
T3	Silvopastoral system under moderate heat stress	5.54 ^a^ ± 1.15	6.15 ^a^	3.01 ^b^ ± 1.12	3.44 ^b^	0.24 ^a^ ± 0.26	0.26 ^a^	0.64 ^c^ ± 0.64	0.71 ^c^	4.23 ^a^ ± 0.83	4.70 ^a^	6.16 ^a^ ± 0.44	6.84 ^a^
T4	Silvopastoral system under intense heat stress	5.5 ^a^ ± 1.04	6.56 ^a^	2.40 ^b^ ± 1.07	3.0 ^b^	0.28 ^a^ ± 0.24	0.35 ^a^	0.22 ^b^ ± 0.57	0.31 ^b^	6.74 ^c^ ± 0.82	7.49 ^c^	9.42 ^d^ ± 0.44	10.47 ^d^

^a,b,c,d^ Averages in the same column with different superscripts differ significantly for the level *p* < 0.05. Feeding behavior (FB); Time in hours (h).

**Table 6 animals-11-01162-t006:** Comparison of the thermoregulatory behavior and other activities of water buffaloes depending on the pastoral system and the season: mean number of animals (standard error) and time in hours.

Treatment	Definition	Lying Down		Standing		Walking		Shading		Wallowing		Thermoregulatory Behavior	
		Number of Animals	Time (h)	Number of Animals	Time (h)	Number of Animals	Time (h)	Number of Animals	Time (h)	Number of Animals	Time (h)	Number of Animals	Time (h)
T1	Conventional system under intense heat stress	2.49 ^a^ ± 0.48	2..24 ^a^	3.63 ^a,b^ ± 0.53	3.26 ^a,b^	4.75 ^a^ ± 0.54	4.27 ^a^	1.90 ^b^ ± 1.02	1.71 ^b^	2.62 ^c^ ± 1.02	2.35 ^c^	4.52 ^c,d^ ± 0.92	4.06 ^c,d^
T2	Conventional system under moderate heat stress	2.10 ^a^ ± 0.46	1.89 ^a^	3.41 ^a^ ± 0.51	3.06 ^a^	4.42 ^a^ ± 0.5	3.97 ^a^	0.68 ^a^ ± 0.94	0.61 ^a^	0.52 ^a^ ± 0.94	0.46 ^a^	1.20 ^a^ ± 0.84	1.08 ^a^
T3	Silvopastoral system under moderate heat stress	2.26 ^a^ ± 0.48	2.03 ^a^	4.07 ^b^ ± 0.5	3.66 ^b^	4.77 ^a^ ± 0.5	4.29 ^a^	2.22 ^b^ ± 1.01	1.99 ^b^	0.77 ^a,b^ ± 1.01	0.69 ^a,b^	2.99 ^b^ ± 0.80	2.91 ^b^
T4	Silvopastoral system under intense heat stress	2.27 ^a^ ± 0.48	2.04 ^a^	3.08 ^a^ ± 0.52	2.77 ^a^	4.44 ^a^ ± 0.53	3.99 ^a^	2.92 ^c^ ± 1.10	2.62 ^c^	1.32 ^b^ ± 1.10	1.18 ^b^	4.24 ^c^ ± 0.97	3.81 ^c^

^a,b,c,d^ Averages in the same column with different superscripts differ significantly for the level *p* < 0.05. Time in hours (h).

## Data Availability

The data presented in this study are available on request from the corresponding author.

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
