# Peer review of "Thermoregulatory and Feeding Behavior under Different Management and Heat Stress Conditions in Heifer Water Buffalo (Bubalus bubalis) in the Tropics"

_animals, 2021, doi:10.3390/ani11041162_

Round 1

Reviewer 1 Report

Thermoregulatory behaviour and its influence on animal productivity remain a topic of some relevance today. However, the present study would be much more original or innovative if some variables had been added or even more innovative methodologies.

Regarding the methodology used to determine the level of thermal stress in which the animals were, without a doubt that THI is one of the most used. However, it is proven that the black globe temperature-humidity index (BGTHI), gives us the perception of the thermal sensation that animals experience since it is considered the effect of the wind. The use of BGTHI would enrich the study and perhaps help to justify some of the results.

In addition, the use of the thermographic camera would have been very useful to justify some of the observed behaviours, which is even mentioned by the authors in the discussion.

Animal production is closely linked to animal productivity and the factors that influence it, such as thermal stress. Thus, it would be important to measure variables or correlate the variables under study with data that would allow us to assess animal productivity (for example, the average daily gain). Since this type of variable was not measured, I suggest that reference be made to productivity and the effects of heat stress on it, when discussing the results (correlating the results with the possible effects they will have on the productivity of the animals).

It is necessary to make a review of the references as some authors are misquoted (wrong names).

Author Response

Reviewer 1

Comments and Suggestions for Authors

Thermoregulatory behaviour and its influence on animal productivity remain a topic of some relevance today. However, the present study would be much more original or innovative if some variables had been added or even more innovative methodologies.

Regarding the methodology used to determine the level of thermal stress in which the animals were, without a doubt that THI is one of the most used. However, it is proven that the black globe temperature-humidity index (BGTHI), gives us the perception of the thermal sensation that animals experience since it is considered the effect of the wind. The use of BGTHI would enrich the study and perhaps help to justify some of the results.

In addition, the use of the thermographic camera would have been very useful to justify some of the observed behaviours, which is even mentioned by the authors in the discussion.

Response: Thank you for your comments. While we did not use the BGTHI nor employed thermographic cameras in this study, we have included this in our recommendations for future studies in lines 324-325.

Animal production is closely linked to animal productivity and the factors that influence it, such as thermal stress. Thus, it would be important to measure variables or correlate the variables under study with data that would allow us to assess animal productivity (for example, the average daily gain). Since this type of variable was not measured, I suggest that reference be made to productivity and the effects of heat stress on it, when discussing the results (correlating the results with the possible effects they will have on the productivity of the animals).

Response: Thank you for your observation, we attempted to address this in lines 313- 317.

It is necessary to make a review of the references as some authors are misquoted (wrong names).

Response: Thank you for this observation, we have reviewed the references and made the necessary adjustments.

Thank you for reviewing our paper and we appreciate the feedback received. We hope that the work that we have done will meet your satisfaction.

Thank you once again.

Reviewer 2 Report

General comments

The manuscript describes the association between behavior and thermal conditions. It is of interest as there is relatively little information on that aspect in buffalo (Bubalus bubalis). The topic of this paper is within the scope of the journal and represents a main research field. The paper is generally well structured, but it is necessary to make some corrections and additions.

Below my indications:

INTRODUCTION

It is well written, describe relevant aspects about the topic to analize

MATERIAL AND METHODS

L 19

ABSTRACT says: “ We used 9 animals with an average weight of 167.9 kg at the beginning of the study” however in line 66 it is not described

L74

Why authors used longitudinal analysis method in the same group of animals.

How could they ensure that the analysis method did not influence the response of the animals, being the same individual?

L 74-75

were the 9 animals selected at the same treatment?

what was the order of selection of each of the 4 treatments?

L 79

I recommend to included the two THI conditions in figure 1.

Indicate with the same symbols as in Table 1, each of the treatments shown in Figure 1.

L 20-21

Abstract says: “…which were observed during the daylight period, from 6:00 to 18:00 hours, at 10 min intervals, for  12 days”   however table 1 (line 80) indicates 9 days. Could you explain this?

L 82-90

I suggest explain with detail how it was assigned treatments, because it is confuse.

Authors selected the month (during May to October), days to introduce the animals where THI values were higher than THI> 75?

How do authors ensure that the effect is due to the treatments or could be the result of a novel stimulus, which would lead to the animals after those three days, to behave differently?

DISCUSSION

I suggest include more updated references to argue the biological aspects found in the research

Ex: Line 207-208

Mota-Rojas, D., Napolitano, F., Braghieri, A., Guerrero-Legarreta, I., Bertoni, A., Martínez-Burnes, J., ... & Orihuela, A. (2020). Thermal biology in river buffalo in the humid tropics: neurophysiological and behavioral responses assessed by infrared thermography. Journal of Animal Behaviour and Biometeorology, 9(1), 0-0.   http://dx.doi.org/10.31893/jabb.21003

Or

Bertoni, A., Mota-Rojas, D., Álvarez-Macias, A., Mora-Medina, P., Guerrero-Legarreta, I., Morales-Canela, A., ... & Martínez-Burnes, J. (2020). Scientific findings related to changes in vascular microcirculation using infrared thermography in the river buffalo. Journal of Animal Behaviour and Biometeorology, 8(4), 288-297.   http://dx.doi.org/10.31893/jabb.20038

REFERENCES

Write the references according to the guidelines of the journal

Ex. Line 443

Reference 43 Year says:  19955

Author Response

Reviewer 2.

General comments

The manuscript describes the association between behavior and thermal conditions. It is of interest as there is relatively little information on that aspect in buffalo (Bubalus bubalis). The topic of this paper is within the scope of the journal and represents a main research field. The paper is generally well structured, but it is necessary to make some corrections and additions.

Below my indications:

Response: Thank you for reviewing our paper and we appreciate the feedback received. We hope that the work that we have done will meet your satisfaction.

INTRODUCTION

It is well written, describe relevant aspects about the topic to analize

MATERIAL AND METHODS

L 19

ABSTRACT says: “ We used 9 animals with an average weight of 167.9 kg at the beginning of the study” however in line 66 it is not described

Response: We thank for your comment. We have added further descriptions in line 66.

L74

Why authors used longitudinal analysis method in the same group of animals.

How could they ensure that the analysis method did not influence the response of the animals, being the same individual?

Response: Thank you for your consideration. We have added the rationale for using the method in lines 76-78 and explained how the analysis method did not influence the response of the animals in lines 82-84. In addition, limitations of the analysis method are noted in lines 210-216.

L 74-75

were the 9 animals selected at the same treatment?

what was the order of selection of each of the 4 treatments?

Response: We thank for your observation. The same animals selected were subject to the all treatments and the order was explained in lines 76-78.

L 79

I recommend to included the two THI conditions in figure 1.

Response: Thank you, we have made it clearer. We inserted THI index to figure 1.

Indicate with the same symbols as in Table 1, each of the treatments shown in Figure 1.

L 20-21

Response: We thank for your observation. We have modified this throughout the document.

Abstract says: “…which were observed during the daylight period, from 6:00 to 18:00 hours, at 10 min intervals, for  12 days”   however table 1 (line 80) indicates 9 days. Could you explain this?

Response: Thank you for finding this error. The number stated in the abstract is correct. Each condition was measured for three days totaling 12 days (Line: 82-84).

L 82-90

I suggest explain with detail how it was assigned treatments, because it is confuse.

Response: We thank for observation. We have done so in lines 77-82.

Authors selected the month (during May to October), days to introduce the animals where THI values were higher than THI> 75?

Response: Thank you for the observation. We have made this clearer in lines 97-99.

How do authors ensure that the effect is due to the treatments or could be the result of a novel stimulus, which would lead to the animals after those three days, to behave differently?

Response: Thank you for the observation. We have explained in lines 82-84 how the previous treatments does not influence the next by the adaptation period employed.

DISCUSSION

I suggest include more updated references to argue the biological aspects found in the research

Ex: Line 207-208

Mota-Rojas, D., Napolitano, F., Braghieri, A., Guerrero-Legarreta, I., Bertoni, A., Martínez-Burnes, J., ... & Orihuela, A. (2020). Thermal biology in river buffalo in the humid tropics: neurophysiological and behavioral responses assessed by infrared thermography. Journal of Animal Behaviour and Biometeorology, 9(1), 0-0.   http://dx.doi.org/10.31893/jabb.21003

Or

Bertoni, A., Mota-Rojas, D., Álvarez-Macias, A., Mora-Medina, P., Guerrero-Legarreta, I., Morales-Canela, A., ... & Martínez-Burnes, J. (2020). Scientific findings related to changes in vascular microcirculation using infrared thermography in the river buffalo. Journal of Animal Behaviour and Biometeorology, 8(4), 288-297.   http://dx.doi.org/10.31893/jabb.20038

Response: We thank for your observation, we have acceded to your suggestion.

REFERENCES

Write the references according to the guidelines of the journal

Ex. Line 443

Reference 43 Year says:  19955

Response: We deleted a number 5 in this cite. Thank you for your observation.
